# Information Entropy Measures for Evaluation of Reliability of Deep Neural Network Results

**DOI:** 10.3390/e25040573

**Published:** 2023-03-27

**Authors:** Elakkat D. Gireesh, Varadaraj P. Gurupur

**Affiliations:** 1Department of Computer Engineering, University of Central Florida, Orlando, FL 32817, USA; 2Epilepsy Center, AdventHealth, Orlando, FL 32803, USA

**Keywords:** certainty, information entropy, iEEG

## Abstract

Deep neural networks (DNN) try to analyze given data, to come up with decisions regarding the inputs. The decision-making process of the DNN model is not entirely transparent. The confidence of the model predictions on new data fed into the network can vary. We address the question of certainty of decision making and adequacy of information capturing by DNN models during this process of decision-making. We introduce a measure called certainty index, which is based on the outputs in the most penultimate layer of DNN. In this approach, we employed iEEG (intracranial electroencephalogram) data to train and test DNN. When arriving at model predictions, the contribution of the entire information content of the input may be important. We explored the relationship between the certainty of DNN predictions and information content of the signal by estimating the sample entropy and using a heatmap of the signal. While it can be assumed that the entire sample must be utilized for arriving at the most appropriate decisions, an evaluation of DNNs from this standpoint has not been reported. We demonstrate that the robustness of the relationship between certainty index with the sample entropy, demonstrated through sample entropy-heatmap correlation, is higher than that with the original signal, indicating that the DNN focuses on information rich regions of the signal to arrive at decisions. Therefore, it can be concluded that the certainty of a decision is related to the DNN’s ability to capture the information in the original signal. Our results indicate that, within its limitations, the certainty index can be used as useful tool in estimating the confidence of predictions. The certainty index appears to be related to how effectively DNN heatmaps captured the information content in the signal.

## 1. Introduction

Previous studies have shown that various information measures can be used for analysis of biomedical signals [1]. The complexity and noise ridden nature of the biomedical signals often make them harder to analyze with standard signal processing tools. This is especially true in the case iEEG, which is a complex signal arising from billions of interactions between neurons, occurring simultaneously. Presurgical evaluation of the patients with epilepsy involves classification of brain regions into epileptogenic or non-epileptogenic based on iEEG signals, so that appropriate clinical recommendations can be made. Information entropy measures including approximate entropy, has been used previously in the classification iEEG signals [2]. Recently, deep learning strategies have been used in classification of the iEEG to that arising from epileptogenic vs. non-epileptogenic regions [3,4] or for identifying interictal epileptiform discharges [5]. In all these scenarios, while classifying data, it is important to capture the entirety of the information present in a signal to arrive at the correct conclusions.

We employ the sample entropy of the signal [6,7] to estimate how well the heatmaps of DNN capture the information content of the signals. The information entropy is a measure of the amount of uncertainty or randomness in a set of data [8,9] and can be effectively used for evaluating the total information, in that system. Ever since Shannon [10] introduced the idea of information entropy, this measure has been employed in estimating the complexity of the signals. Later, approximate entropy was introduced which in addition takes into account the regularity in the data, employing the idea of pseudo-phase [11]. A larger value of the approximate entropy corresponds to higher complexity of the signal along with low number of repeated patterns. While approximate entropy helps in evaluating the nature of the data generating system, it does heavily depend on the length of the records and can underestimate for shorter signal lengths. Subsequently, sample entropy was introduced which excludes the self-matches, which in the case of approximate entropy introduce a bias suggesting more regularity than reality. Moreover, sample entropy has comparatively reduced dependence on the signal length and has higher relative consistency [1,6].

While both approximate entropy and sample entropy has been used in analysis of neural respiratory signals [12] the sample entropy showed more consistent results. Further, exploration of the various embedding dimension and data lengths of phrenic nerve discharge data in this study revealed the critical role of exploring the parameter space in both types of entropy measurements. Another study which explored the role of embedding a dimension in the calculation of various forms of entropy in the cardiac signal also demonstrated that the results can vary based on the embedding dimension [12]. Keeping these concerns in mind, in this study, we have explored the parameter space for calculation of sample entropy.

In the field of deep learning, the question of the relevance of data captured by the DNN, is addressed through evaluation of heatmaps. Once a DNN model is trained on a dataset, the reliability of information capturing capacity of DNN can be assessed by evaluating heatmaps, which help in explaining what the DNN has identified as significant components of the data, in arriving at a particular decision. Heatmaps are a method for visualizing which specific parts of a signal or image are most relevant for a prediction made by the model. They are generated by analyzing the activations of the DNN, and in the case of convolutional neural networks, typically the activations of last convolutional layer, which is presumed to contain the most abstract and high level features of the input, that were most important for the DNN’s decision [13]. For this purpose, heatmaps can be calculated using different methods, which includes, 1. sensitivity analysis [14], 2. simple Taylor decomposition [15], 3. layer-wise relevance propagation [16], and 4. gradient class activation maps (Grad-CAM) [17]. The method of Grad-CAM has been effectively used in visualization of the relevant regions of the images [18] or signals [19,20] that lead to the decisions of DNN. In the context of this study, heatmaps were calculated using Grad-CAM for evaluating the iEEG signal, which identified the important regions that contributed to the decision of whether the signal belong to either epileptogenic or non-epileptogenic class.

Another aspect of deep learning involves the uncertainty related to individual predictions. Given any trained DNN model, the prediction of any newly presented data will be associated with some level of uncertainty. These types of uncertainty include, uncertainty caused by model, data uncertainty, or distributional uncertainty [21]. Model (epistemic) uncertainty can be reduced by increasing size of training data. On the other hand, data uncertainty or aleatoric uncertainty can be irreducible as it is related to the complexity of the data. Distributional uncertainty is related to mismatch between the training and test distributions. The use of softmax measures has been described, especially as a measure of epistemic uncertainty (certainty measure against samples from out of distribution data) [22]. While this study explored the possibility of using softmax confidence as a proxy of epistemic uncertainty, they do stress that it is an imperfect measure.

In general, uncertainty approaches try to give an overall estimate of the uncertainty of predictions. While they are helpful in estimating the confidence one can have in the outcomes of the model, there is no specific score on the reliability of an individual prediction, for example, on the classification of a particular iEEG signal. We introduce a score, for the reliability of an individual prediction, called certainty index. It is an index measuring how certain the given model is about a specific classification. This certainty of the prediction is assessed through the differences in the outputs prior to softmax layer. This method was motivated by the following two ideas: 1. In the case of biological neural networks, the decision of a subsequent neuronal firing is partly dependent on the summation of the post-synaptic action potentials (both excitatory and inhibitory), which is similar to the inputs to DNN’s last layer. Broadly, we can consider the “decision” of the neuron to fire and transmit the information to the next layer as a surrogate of certainty. Therefore, intuitively, we can consider that the biological neurons are considering the positive and negative inputs in arriving at the decision and possibly at assessing the certainty of predictions. 2. The Grad-CAM algorithm used in the heatmap generation is based on the gradients of scores for individual classes (before the softmax layer) [17], explained in detail in the methods section. Therefore, using a certainty measure based on the same score was considered appropriate when evaluating the relationship between heatmap and information measures such as sample entropy used in this study.

Taking the approaches of information estimation, as noted above (sample entropy), this work aims to establish the relationship between certainty of decision of DNN and efficacy of model in capturing the information content in the signal. The regions of heatmap with higher values is considered to be the more relevant regions of the signal, from where model captured important information. We were able to demonstrate that the certainty of the predictions of trained DNN model is directly proportional to how well the DNN captures the information content in the signal.

### Summary of Contributions

Exploring the information processing in the DNN, with the help of sample entropy, we addressed the following.

How to use the outputs of the DNN (at penultimate layers) as a measure of certainty of individual decisions.Evaluate the correlation between heatmap of DNN and sample entropy of the signalExamine relationship between heatmap-sample entropy correlation and the certainty of individual decision.

In this study we introduce the measure of certainty index which estimates the confidence of an individual prediction by a DNN model. We establish a method for correlating sample entropy signal with the heatmap and estimate that a high correlation between these two time series indicates a higher certainty of the DNN prediction. This result was noted to be valid even with changes in the parameters of the estimation of sample entropy.

## 2. Materials and Methods

This study was based on intracranial EEG data acquired from patients undergoing monitoring for possible epilepsy surgery. Briefly, intracranial EEG data from 10 patients undergoing epilepsy surgery evaluation were recorded using intracranial electrodes, continuously for several days using Nihon Kohden (NK) software at a sampling rate of 2000 Hz. The electrode placement was decided based on pre-surgical evaluation. For the purpose of analysis in this study, data of one minute duration from interictal periods (when the patients did not have any identified seizures) were used. Signals were filtered between 60–600 Hz, processed in bipolar montage, and exported to the EDF format from NK software. The deep neural network implemented using Tensorflow package from Google. Inc., Mountain View, CA, USA, shown in Figure 1A was trained with one second data at a time. The data from all the 10 patients was shuffled before training the network, to minimize the bias due to contribution from individual patients. The DNN was used for classification of the data to epileptogenic vs. non-epileptogenic zone. After the model was developed with the initial data, the model was retested using data from 7 additional patients, acquired in the same way as noted above. The true values on whether an electrode location belong to epileptogenic or non-epileptogenic was decided based on successful surgery outcome. The data was processed using high performance computing facility with GPUs. The study was approved by Adventhealth Orlando IRB.

### 2.1. Estimation of Uncertainty

As noted in the introduction, the certainty of model predictions has been a major challenge in the field of deep learning. We introduce a method of assessment of certainty in the neural network predictions.

This is a measure of how certain, DNN is about each of the individual predictions. This is different from accuracy which is a measure of overall accurate predictions. The certainty in the outputs of *n*-th layer is estimated as follows:(1)$$C_{i}^{n}=y_{i}^{n}-\frac{1}{N}\sum_{j=I;j\neq i}^{N-1}y_{j}^{n}$$
where certainty index, $C_{i}^{n}$ is the certainty that the *i*-th prediction in the *n*-th layer is correct, $y_{i}^{n}$ is the *i*-th DNN output at *n*-th layer, *N* is the total number of nodes. In this study, the *n*-th layer is chosen to be the penultimate layer (immediately prior to the softmax). The second summation term indicates the average of all the other DNN predictions other than the *i*-th DNN output. For comparison between layers and different DNN’s certainty value can be normalized to standard deviation. This will provide an individual value for each of the sample signals in the EEG data.

Since the total number of nodes in the current study for classification is only two, (epileptogenic vs.. non-epileptogenic), the certainty index calculated based on the outputs in the penultimate layer will be equal to: *y_i_* − *y_j_*, which is used in the subsequent analysis.

### 2.2. Heatmap Estimation

Different methods have been described for the estimation of heatmap of a DNN prediction, and Grad-CAM has been commonly applied in the case of signals. Class-discriminator map defined as Grad-CAM is evaluated for any class *c*, as follows [17]. The gradient of the DNN output score, *y^c^* for class *c*, with respect to the feature map activation *A^k^* of a convolutional layer is given by $\frac{\partial y^{C}}{\partial A_{i}^{k}}$. This value is calculated through successive matrix products of weight matrices and gradient with respect to activation functions. The GradientTape algorithm of Tensorflow was used to record the operations (activation functions) in forward flow, which was later used in performing automatic differentiation to calculate the gradients. The *k* in this case, indexes the different feature maps of a particular convolutional layer. The *i* indexes the individual elements in the feature map, given the fact that the feature maps are obtained by one dimensional convolution. These gradients can be global-average pooled, which gives weights (*α^c^*) of neurons based on their significance in decision making for a specific input [17,19].
(2)$$\alpha_{k}^{C}=\frac{1}{N}\sum_{i=1}^{N}\frac{\partial y^{C}}{\partial A_{i}^{k}}$$
$y^{C}$ is the DNN output for a particular class *c (before softmax)*.$\alpha_{k}^{C}$ indicates the importance weight of *k*-the filter for class *c*.$A_{i}^{k}$ is the *i*-the element in *k*-th activation map.*N* is the number of elements in feature map.

With the gradient weighted class activation map (grad-CAM) for a layer obtained as
(3)$$g^{c}=\Sigma_{k}\alpha_{k}^{C}A^{k}$$

The ReLU function was not used, given the fact that we are working on one dimensional signal and similar approach was used in acoustic signal based studies [19]. Also, incorporating negative values in heatmaps was considered to be important when calculating correlations with information measures of the signal. This approach helps in generating the heatmap for any layer. Since the concern that the Grad-CAM maps can progressively worsen in the earlier layers, in this study, the heatmap was only calculated based on the last convolutional layer of the model.

To estimate the similarity between the heatmap (*g^c^*) and the original data *X* (with a series of *x*_1_*, …, x_n_*), a cross correlation was performed as
(4)$$zk=\sum_{l=0}^{g^{C}-1}g_{l}^{C}*x_{l-k+N-1}^{*}$$
where ||*g^c^*|| is the length of *g^c^*, which is the heatmap for signal *X*; *N*= max (||*g^c^*||,||*X*||) and $x_{m}^{\;}$ is 0 when *m* is outside the range of *X*. The maximal values of the correlations (max (*z*)) were used for plotting against the certainty index.

### 2.3. Sample Entropy Calculation

The information in a collection of data *X* can be defined as
(5)$$H\left(x\right)=-\sum_{x\in x}p\left(x\right)\log p\left(x\right)$$
where *X* is taking values *x*_1_*, …, x_n_* and *p(x)* is the probability associated with those values for all *x*_1_*, …, x_n_*. Compared to a random set of data, the iEEG signals are characterized by repeatable patterns which also carry a component of information content of the signal. That would suggest that a regular entropy estimation may not appropriately capture the information content in a time series like iEEG. To address this concern, the sample entropy in the individual data was calculated as follows. Given a sequence of numbers *x*_1_, *x*_2_, …, *x_n_* of length N, a non-negative integer m ≤ N and a positive integer r, block *u(i)* can be defined as *x(i), x(I + 1)…, x(i + m − 1*) and block *u(j) as x(j), x(j + 1)…, x(j + m − 1*). The distance between them is defined as *d[u(i), u(j)] = max_k=1,2,…m_(|x(i + k − 1) − x(j + k − 1)|).*

The sample entropy, which helps in better capturing the recurring nature of data elements in a signal, is defined as [7]
(6)$$SampEn\left(m,r,N\right)=-\log\frac{\sum_{i=1}^{N-m}\sum_{i=1,j\neq i}^{N-m}\;}{\sum_{i=1}^{N-m}\sum_{i=1,\;j\neq i}^{N-m}\;}\frac{\left[number\;of\;times\;that\;d\left[\left|u_{m+1}\left(j\right)-u_{m+1}\left(i\right)\right|\right]<r\right]}{\left[number\;of\;times\;that\;d\left[\left|u_{m}\left(j\right)-u_{m}\left(i\right)\right|\right]<r^{\;}\right]}$$
where *m* represents the embedding dimension based on which the data is split, *r* the scaling parameter, usually measured as multiples of standard deviations of the signal, and *N* the length of sample considered. For signal of length *n*, the sample entropy series SE is calculated for *n* moving windows of size N to produce a linear vector. The algorithm used for implementation of the sample entropy for a time series is shown in Algorithm 1. The cross correlation between the heatmap and sample entropy is calculated similar to the method described in Equation (4), with the original data (*x*) replaced by SE, a series of the same length as the original signal.
**Algorithm 1** Sample Entropy for a time seriesSample entropy of signal *s* of length SN for embedding dimension *m*, scaling parameters d and sample entropy calculation length *N***Input:** s1, s2,…..s_SN_**Output:** SE_1_, SE_2_… SE_N_ (series of sample entropy)1: SE ← [0_1_, ------0_SN_]2: N ← length(s)3: m ← embedding dimension4: r ← scaling parameter5: for <si in range of SN> do sig_si_ ←<split s into SN segments of length N> end for6:   for <*i* in range of *N-m*> do *xmi*←<split *sig_i_* into segments of length m> end for7:   for <*i* in range of *N-m+*1> do xmj← <split *sig_i_* into segments of length m> end for8:    B←<total of the modulus (xmi-xmj) <r> [xmi-xmj indicates the distances between the segments]9:   m←m+110:    for <*i* in range of *N-m*+1> do xmk ←<split *sig_i_* into segments of length m> end for11:     A← <total of the modulus (xmk-xmk)> [xmk-xmk indicates the distances between segments]12: SE_si_ ← −log (A/B)

The maximal value of cross correlation value was plotted against the certainty values for the same data obtained through DNN model. The R-squared values and F-values for each these plots were estimated with regression analysis and tabulated. To further validate the robustness of the results, sample entropy for each signal was calculated at various embedding dimensions (m = 4, 8, 16, 32), scaling parameters (r = 1.5, 2, 2.5) and sample lengths (50, 100, 200, 400).

### 2.4. k-Fold cross Validation

A ten-fold cross-validation was employed for assessing the consistency of accuracy of the model [23]. Initially, the iEEG data were randomly divided into ten equal portions. Nine out of ten portions of iEEG signals were used to train the DNN and the remaining one-tenth of the iEEG signals was used to test the model. The above strategy is repeated ten times by shifting the test and training dataset. The average accuracy along with the standard error was reported.

## 3. Results

### 3.1. Model Accuracy

A convolutional network model was implemented as described in Figure 2. The model consisted of three convolutional layers and additional dense and dropout layers. A dense layer was added before the softmax layer to get the outputs prior to the softmax function. The iEEG data lasting one second from each channel were fed into the input layer. The model was trained with 9000 samples and tested with 1000 samples. The accuracy of the model was noted to be 93%. The confusion matrix which depicts the percentage of positive and negative predictions is shown in the Figure 2B.

The model was further validated with a 10-fold cross validation, with the data split into 10 separate folds. This yielded an accuracy score of: 91 ± 0.2% (standard error). To further validate this model, additional data from seven patients was evaluated with same model architecture and a similar 10-fold analysis was performed on that data. A similar accuracy of 95 ± 1% was noted in this analysis.

### 3.2. Certainty in Individual Predictions

The confidence in the prediction of each data element was estimated as noted by the measure of certainty as described in the methods section. The absolute value of certainty, for signals from non-epileptic electrodes ranged between 0 and 200 and that for the epileptic electrodes ranged from 0 to 60 (for representational purpose in figure, the more negative the value, the higher the certainty that the data is from epileptic regions) (Figure 2). For better comparison, the data were also plotted after normalizing with the standard deviation.

### 3.3. Certainty and Correlation between Heatmap and Signal

To estimate how the decision-making process of the DNN model is related to the certainty index, we calculated the correlation between the heatmap and the original signal (using Equation (4)). The maximal correlation values were plotted against the certainty index for that signal (Figure 3). The R-square values are given in the Table 1. A similar range of correlation value was obtained when the epileptic and non-epileptic data were evaluated separately.

### 3.4. Relationship between Heatmap and Sample Entropy of Signal

The sample entropy was calculated based on the Equation (6). For this purpose, following parameters were used: embedding dimension, m = 8, scaling parameter, r = 2 × standard deviation of the signal, signal length, N = 100. The fact that the EEG signal has frequency components which range from 60 to 600 hz was considered in choosing the embedding dimensions and signal length. Further analysis based on variations in these parameters are noted in sections below. A cross-correlation was calculated between the sample entropy and the heatmap and the maximal value of this cross correlation was plotted against the certainty index for individual data as shown in Figure 4. The R-squared values for the regression analyses are given in the Table 2. It may be noted that the R-squared values in the case of the cross correlations between the sample entropy and heatmap appear to be higher compared to that between original signal and heatmap.

### 3.5. Relationship between Heatmap and Sample Entropy at Various Embedding Dimensions

The Equation (6) shows that the sample entropy depends on the embedding dimension *m*, scaling parameter *r*, and signal length *N*. A too high value of *m* can potentially reduce the template matches performed in the algorithm. On the other hand, if *m* selected is too small, there will be more template matches but the predictive information will be reduced, and the probability of forward match can be underestimated. This is especially true in the case of EEG which may have repeating patterns. To evaluate the impact of these parameters on the relationship between sample entropy, heatmap, and certainty index, those parameters were varied, and the relationship was estimated. The sample entropy, m, was calculated at embedding dimensions 4, 8, 16, and 32 (while keeping r = 2, N =100). The certainty index vs. maximal correlation between the sample entropy and heatmap at various embedding dimensions is plotted in Figure 5. The corresponding regression values are presented in Table 3.

### 3.6. Sample Entropy Calculated at Different Standard Deviations

Similarly, if a high scaling parameter (*r*) value is selected, most of the templates will look like each other and they will fall below threshold and therefore the algorithm will have reduced efficiency. If *r* is too small, too many templates will fail to match. To address the effect of these variations, the sample entropy was calculated at various *r* values while keeping *m* = 8 and *N* = 100. Plots for the variations in the standard deviations (at 1.5, 2, and 2.5) are shown in Figure 6 and corresponding regression values are noted in Table 4.

### 3.7. Sample Entropy Calculated at Different Sample Lengths

Another parameter considered in the calculation of sample entropy is length of the signal (*N*). This is the moving window of the original signal. Given the different frequency components in the EEG signal which will span different lengths of signal for each frequency, the signal length *N* was varied to calculate the sample entropy. Sample entropy was calculated at various N values while keeping m = 8 and r = 2. Plots for the variations in the standard deviations (at 1.5, 2 and 2.5) are shown in Figure 7 along with the corresponding regression values in Table 5.

A high correlation (as noted in regression-based R-squared- values) between the certainty index and the maximal correlation between sample entropy and heatmap continue to be present in the various parameter spaces considered.

## 4. Discussion

Our study evaluated how the ability of the DNN model to capture the information content of the signal influences the certainty about the predictions. For this purpose, we introduced certainty index as a measure of confidence of individual DNN predictions, which is based on the outcomes prior to softmax layer. Given the fact that the heatmaps towards the final convolutional layers are representative of the highest-level abstractions of the information in the signal that the DNN is using to arrive at the decisions about sample classes, we used a measure of the correlation between that heatmap in the final convolutional layer and original signal to evaluate how effectively the DNN captured the information in the signal. Sample entropy measure was used as measure of information content in the signal (which was calculated independent of the DNN model), and we were able to demonstrate that the certainty index of each sample is proportional to the correlation between sample entropy and heatmap.

We would like to point out that the model that we developed for this purpose was comparable to the previous DNN models reported using intracranial EEG data (noted in Table 6). As noted previously, the main aim of this study was to explore the certainty of predictions of DNN with the help of heatmap and information content of the signal. The fact that the model we used was comparable to the reported models in accuracy suggests a potential applicability of the approaches currently employed in other models. It may be noted that the seizure detection studies usually demonstrate higher accuracy compared to the classification of data to epileptogenic or non-epileptogenic classes from inter-ictal periods (time periods when there was no seizures recorded).

### 4.1. Certainty Index as a Measure of Confidence in Individual Decision of DNN

Confidence measures for DNN predictions has been evaluated in the past, particularly using outputs of softmax layer. Logits from the softmax layers give a range of values that appears to give a confidence of predictions, but previous studies have also cautioned that this can be erroneous [28], especially given the discontinuous nature of input-output mappings, and should be used judiciously. Softmax confidence was further evaluated and has been considered an imperfect measure of uncertainty [22], especially for evaluating epistemic uncertainty. That study analyzed the softmax function and defined regions of softmax layers where an out of distribution input must fall to be correctly labelled as out of distribution. Statistics derived from softmax distributions were effective in determining whether a sample is misclassified or from a different distribution from the training data, suggesting its potential as a measure of certainty [29]. This study showed potential applications of this approach in diverse experimental data, including computer vision and natural language processing.

The method of certainty measurement that we are proposing, Incorporate the scores of the network (with values prior to the softmax layer), that favor a particular prediction and un-favor the other options, further enhancing the reliability of this measure. It may be noted that limitations attributed to the softmax based confidence measure may be present in this approach as well, but the improvisation incorporating the negative prediction outputs will hopefully make it more robust. As discussed in the introduction, we preferred to use the values prior to the softmax layer for certainty index calculation, taking cues from biological neural networks. The other reason was that, given the approach of Grad-CAM which used the gradients of DNN outputs, prior to the softmax layer, estimation of certainty based on those values appeared more appropriate. While this may improve the reliability of this measure, further studies with different datasets may be needed for further ascertaining the wider applicability of this approach.

While the measures of confidence have been addressed in various ways (Table 7), a rigorous evaluation of this measure from the standpoint of heatmaps and information content of the samples has not been reported. Our main objective in this study was to establish how strongly this kind of confidence measure relates to the measures of information content in the signal. To the best of our knowledge, this is the first study addressing the relationship between confidence of prediction (as measured by certainty index), heatmap, and information content of the signal.

### 4.2. Relationship between Heatmap and Original Signal

Heatmaps calculated using various algorithms demonstrate the focal regions in the signal which the DNN identified for arriving at the decisions. By estimating the correlation between the heatmap and original signal, we aimed at capturing how much the features of the original signal match with the heatmap. The fact that the convolutional layers generate the heatmaps which specifically pay attention to specific regions of the signal suggests that there may be regions of the signal that do not significantly contribute to the decisions.

### 4.3. Use of Correlation between Heatmap and Sample Entropy

To estimate the information content of the signal, we used the method of sample entropy, modified in this study, for time series. A higher correlation coefficient in the case of sample entropy and heatmap vs. certainty index suggests that the information captured with the sample entropy is significant in the decision making of DNN.

It may be noted that the sample entropy is calculated from the original signal with a completely independent algorithm (than DNN or heatmap generation), which helps in capturing the information content in the signal. Sample entropy has been studied in the past to assess the regularity in the iEEG signal [33]. This study addressed the variations in the sample entropy in sleep states, demonstrating that the information in iEEG signals can be captured using sample entropy. Another study with vibration signals employed sample entropy based DNN to successfully diagnose faults in automobile systems [34]. It may be noted that this study did not assess the DNN based on sample entropy, but rather used it to modify the input to DNN to improve the outcome. A regional perturbation based method of assessing a heatmap was employed [13] which can compare the various heatmap performances. While this method allows for assessing which heatmap is most informative, it did not directly address the question of how effectively the heatmap is capturing data in the input signal.

We have tried to correlate the reliability of DNN predictions to one of the fundamental features of the signals- the information content, by using sample entropy. We have demonstrated that the relationship between the heatmap and sample entropy is better correlated with the certainty of predictions of DNN. The fact that the certainty index shows a higher R-squared statistic with heatmap–sample entropy correlation compared to heatmap–original signal correlation suggests that the sample entropy may be a potential tool in assessing whether the DNN model is capable of capturing the entire information in the signal.

### 4.4. Variations in the Estimation Parameters Demonstrated Robust Relation between Certainty Index and Correlations

To further evaluate the robustness of the above-described relationships, further analysis was carried out across the parameter space of sample entropy calculations. Previous studies have shown that the estimation of sample entropy can be affected by variations of the embedding dimension (m), scaling parameter (*r*), and sample length (*N*) [12]. We were able to show that the trend of the correlation between the sample entropy-heatmap and the certainty index will remain robust even at various parameter spaces. In this study, the variation of sample entropy was estimated in the range 8–32 for embedding dimensions, and 50–400 for sample length, both in a logarithmically increasing size, to maximize variability.

### 4.5. Limitations

The certainty index that we introduced, cannot be taken as a perfect measure of confidence. Small variations in the signal could change this index significantly and therefore should be corroborated with the rest of the data. Our study did not rigorously examine the use of certainty index on all types of networks or data. In this study, the main purpose of certainty index was to have a measure with reasonable confidence to correlate with the information content in the signal. Moreover, our study did not evaluate out of distribution examples.

It may be noted that the certainty measure is not a general estimate of the confidence of the network, but rather an estimate in the individual predictions. It is possible that the average of certainty measure over multiple samples of data may be used as a mean certainty. 

We have not explored additional information measures, which may have potential applications or better outcomes in terms of correlation with information parameters. In addition, the data we used were limited to one type of signal (iEEG), which raises a question concerning the applicability of this approach to other kinds of data.

### 4.6. Future Studies

The certainty index introduced may be evaluated for different types of networks (RNN, LSTM etc.) and different types of datasets (e.g., images). Other measures of information, e.g., fuzzy entropy, can be used as an alternative to sample entropy to evaluate the certainty index further. Another aspect that needs to be explored is the relationship of the gradients of the network with the certainty index. This will potentially help in deciding the optimal number of layers in a network.

## 5. Conclusions

Exploring the information capturing capacity of the DNN with the help of heatmapping and sample entropy in iEEG signals, we were able make the following contributions.

Introduced *certainty index* as a measure of confidence of individual predictions of DNN using the outputs of the penultimate layers.Modified an algorithm for measuring sample entropy of a time series.Evaluated the relationship between the heatmap–sample entropy correlation and the certainty index.Established that the trend of relation between the certainty index and heatmap-sample entropy correlation is robust across different parameters of sample entropy calculation. 

The results from this study suggest that the certainty measure as described can be used judiciously as an estimate of the confidence that the DNN has in the prediction of particular data. We are able to show that the certainty index is more strongly correlated with the correlation between the heatmap and sample entropy compared to that between heatmap and original signal. This would be expected given the fact that the sample entropy is the measure of the information in the signal and the heatmap is a depiction of the DNN’s ability to capture the relevant information.

## Figures and Tables

**Figure 1 entropy-25-00573-f001:**
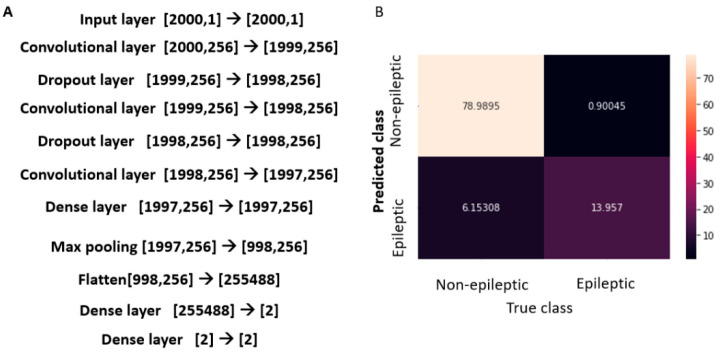
DNN model: The schematic of the model along with the confusion matrix which gives an estimate of the accuracy of predictions. (**A**) The scheme of the model used in the study, the input and output sizes are shown in the brackets. (**B**) Confusion matrix for the test samples (1000) represented as percentages with predictions by the model.

**Figure 2 entropy-25-00573-f002:**
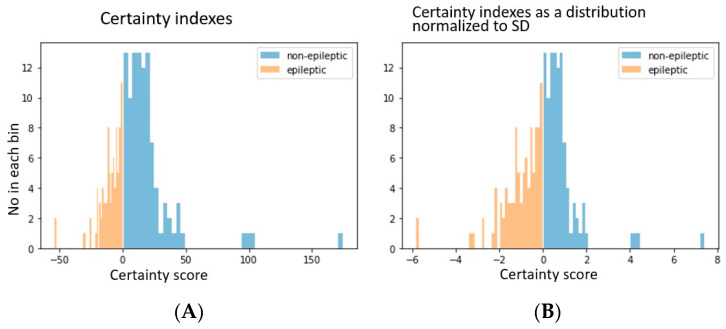
The certainty indexes in classification of data to epileptic or non-epileptic locations. (**A**). The actual certainty index values plotted as histogram. The negative value for the epileptic group is given for demonstration purpose only (**B**). The certainty indexes normalized to their standard deviation (separate standard deviation for either class).

**Figure 3 entropy-25-00573-f003:**
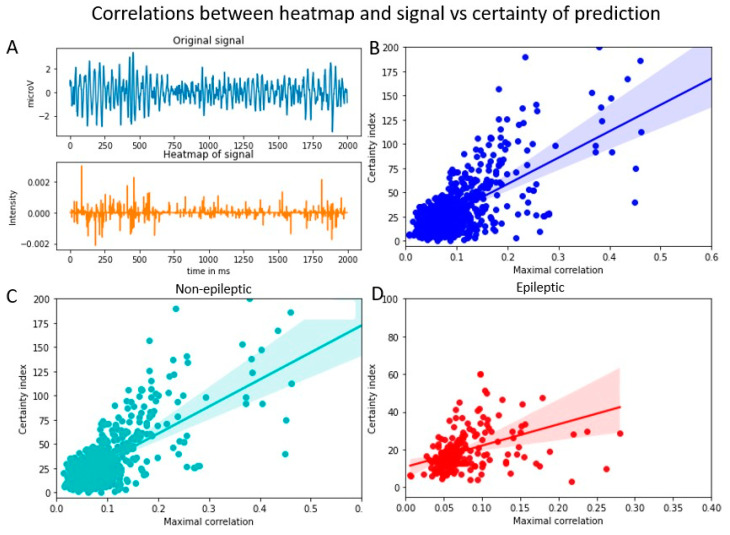
The certainty index and its relationship with the correlation between heatmap and original signal. (**A**). Original signal (**upper**) and the corresponding heatmap (**lower**) (**B**). Scatter plot: Certainty index vs. correlation between heatmap and original signal (**C**). Cross-correlation between the heatmap and original signal plotted for non-epileptic data. (**D**). Cross-correlation between the heatmap and original signal plotted for epileptic data.

**Figure 4 entropy-25-00573-f004:**
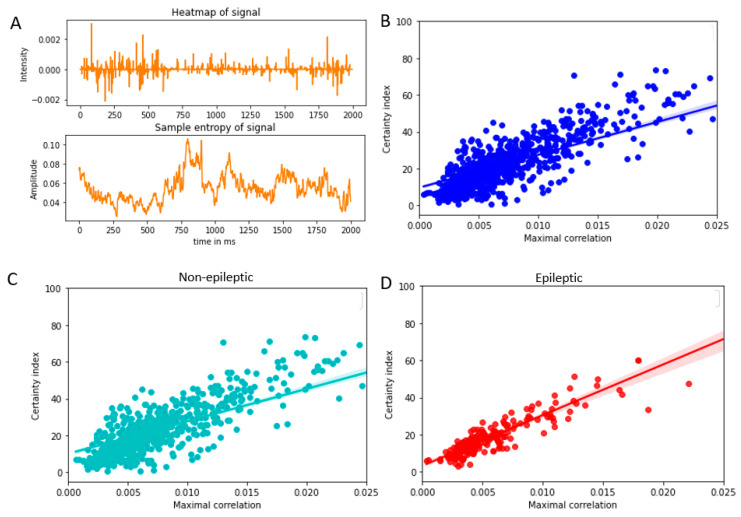
The certainty index and its relationship with the correlation between heatmap and sample entropy of the signal. (**A**). Heatmap of the signal (**upper**) and the sample entropy (**lower**) (**B**). Scatter plot: Certainty index vs. correlation between heatmap and sample entropy of signal (**C**). Cross-correlation between the heatmap and sample entropy of signal for non-epileptic data. (**D**). Cross-correlation between the heatmap and sample entropy of signal plotted for epileptic data.

**Figure 5 entropy-25-00573-f005:**
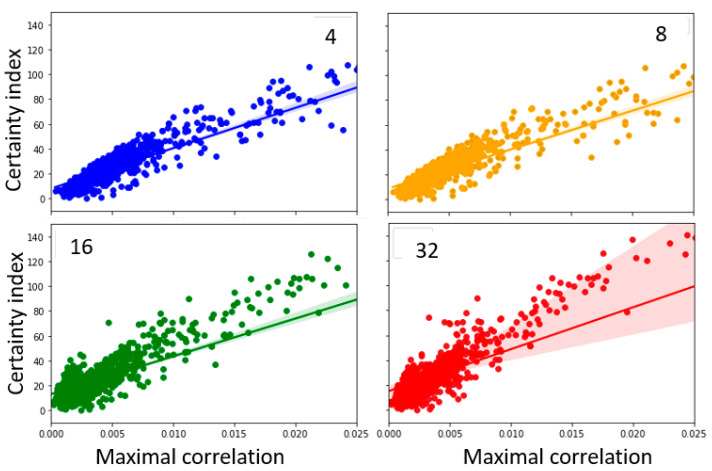
Sample entropy calculated for varying embedding dimensions: 4, 8, 16 and 32. The labels for the figures correspond to the individual embedding dimensions.

**Figure 6 entropy-25-00573-f006:**
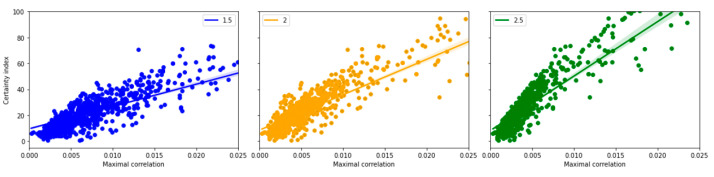
Sample entropy calculated for different scaling parameters where (r =1.5, 2 and 2.5 × SD). The labels for the figures correspond to the individual scaling parameter × SD.

**Figure 7 entropy-25-00573-f007:**
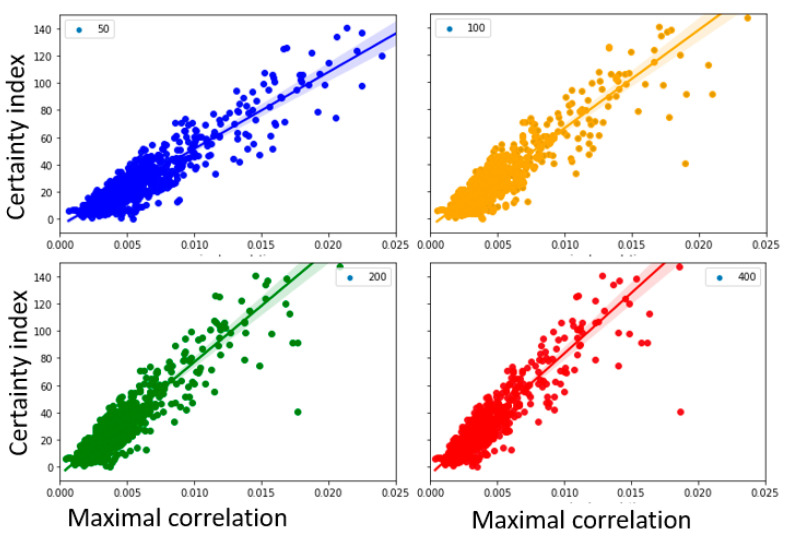
Sample entropy calculated for different signal lengths: 50, 100, 200 and 400. The labels for the figures correspond to the individual signal lengths.

**Table 1 entropy-25-00573-t001:** Regression values: certainty index vs. correlation between heatmap and original signal.

Statistical Values	Whole Data	Non-Epileptic	Epileptic
R-squared	0.76	0.77	0.69
F-statistic	3190	2714	450

**Table 2 entropy-25-00573-t002:** Regression values: certainty index vs. correlation between heatmap and sample entropy.

Statistical Values	Whole Data	Non-Epileptic	Epileptic
R-squared	0.89	0.90	0.95
F-statistic	8303	6850	4197

**Table 3 entropy-25-00573-t003:** Regression values: certainty index vs. correlation between heatmap and sample entropy at various embedding dimensions.

Statistical Values	4	8	16	32
R-squared	0.91	0.91	0.83	0.63
F-statistic	11,333	10,402	5201	1705

**Table 4 entropy-25-00573-t004:** Regression values: certainty index vs. correlation between heatmap and sample entropy at various scaling parameter (SD).

Statistical Values	1.5	2	2.5
R-squared	0.93	0.93	0.93
F-statistic	14,016	13,796	14,346

**Table 5 entropy-25-00573-t005:** Regression values: certainty index vs. correlation between heatmap and sample entropy at various signal lengths.

Statistical Values	50	100	200	400
R-squared	0.93	0.93	0.93	0.93
F-statistic	13,509	13,796	14,039	13,611

**Table 6 entropy-25-00573-t006:** Existing DNN models on iEEG, exploring epileptogenic zones/ epileptic activity compared with the current model.

Study	Deep Learning Strategy	Input Formulation	Frequency Range (FR)/Sampling Rate(SR)	Task	Accuracy
[3]	CNN(Convolutional neural network) with STFT(short term Fourier transform)	Data from 5 patients. 20 s of data	SR: 512 Hz	Differentiate focal and non-focal epileptogenic signal	91.8%
[4]	1D-CNN with data augmentation strategies	24 patients, 916 h data; & 18 patients,2565 h data.	SR:256 Hz	Seizure detection	99%
[5]	CNN	Data from 12 patients.	NA (not available)	Interictal epileptic discharge detection	79–87%
[24]	CNN + LSTM (long short-term memory attention machine)	Three data samples	SR:1–512 Hz, 1–173 Hz or 2048 Hz	Epileptogenic vs. non-epileptogenic	97.6%
[25]	1-CNN, 2-CNN, 3-CNN, 4-CNN	2016 Kaggle competition; Data from 5 dogs and 2 patients	SR: 400 Hz	Seizure classification	76–95%
[26]	CNN	2016 Kaggle competition; Data from 5 dogs and 2 patients	SR: 400 Hz	Seizure prediction	87.85% sensitivity in seizure prediction
[27]	CNN	Responsive neural stimulator data from 22 patients	SR: 250 Hz FR: 4–125 Hz	Seizure identification	84%
Current study	CNN	Data from 17 patients; 1 min data	SR: 2000 Hz; FR:60–600 Hz	Epileptogenic vs. non-epileptogenic	91–95%

**Table 7 entropy-25-00573-t007:** Existing literature compared with our approach for evaluation of the certainty of the network and assessment of the certainty measure.

Publication	Employed Method	Assessment of the Confidence Measures	Comparison to Our Approach
Hendrycks et al. [29]	Softmax prediction probability	Correctly classified examples tend to have greater maximum softmax probabilities	Did not assess for the relationship between information content in the samples.
Jha et al. [30]	Attribution based confidence measure	Studied effect of changing the labels of features away from the sample studied and conformance of model predictions.	Established attribution based. dimensionality reduction
Smith et al. [31]	Mutual information and softmax variance	Mutual information, expected Kullback-Leibler Divergence and predictive variance help in computing the divergence between softmax and expected softmax.	Considered softmax variance as a measure of mutual information
Pearce et al. [22]	Analytically studied softmax layer	Studied the effectiveness of softmax outputs as proxy for epistemic uncertainty in non-adversarial, out of distribution examples	Suggested partial capture of uncertainty. Did not not explore relationship with heatmaps or information content of samples
Lakshminarayanan et al. [32]	Ensembles of neural networks	Used 1. scoring system as training criterion, 2. adversarial training	Evaluated entropy of predictive distributions to evaluate quality of uncertainty estimates; Evaluated performance compared to Baysian networks

## Data Availability

Available on request.

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
