# Peer review of "Information Entropy Measures for Evaluation of Reliability of Deep Neural Network Results"

_entropy, 2023, doi:10.3390/e25040573_

Round 1

Reviewer 1 Report

The authors introduce a certainty index to measure the confidence of the predictions of a neural network. For that they worked on an existing sample entropy algorithm adjusted to the problem and evaluated the relationship between heatmap-sample entropy correlation and the proposed certainty index. The application domain of the experiments regards EEG signals including epileptic and non-epileptic data.

- In equation (1) some symbols are not explained.
- In equation (2) it is not clear how the derivative of y is estimated.
- It is advised to apply the proposed method in more data.
- The experiments don't include direct comparison with other methods to quantitatively show why the proposed method is better.
- The manuscript is in general well written, but the introduction and the related work need more work.

Author Response

Response to Reviewer 1

Comment 1.1: The authors introduce a certainty index to measure the confidence of the predictions of a neural network. For that they worked on an existing sample entropy algorithm adjusted to the problem and evaluated the relationship between heatmap-sample entropy correlation and the proposed certainty index. The application domain of the experiments regards EEG signals including epileptic and non-epileptic data.

Response 1.1: Thank you very much for the valuable comments from the reviewers which has significantly helped in improving the manuscript. We have made the appropriate changes in the main text.

Comment 1.2: In equation (1) some symbols are not explained.

Response 1.2: We have included detailed explanation of the symbols in equation (1)

Comment 1.3: In equation (2) it is not clear how the derivative of y is estimated.

Response 1.3: The calculation of the gradient with the help of activation functions and weight matrices is described in section. The use of GradientTape algorithm, from tensor flow was incorporated.

Comment 1.4:
It is advised to apply the proposed method in more data.

Response 1.4: Additional patient data is included and added in section 3.1. Also, as recommended by the other reviewer we have included a 10-fold cross validation of the results of initial data and the additional data.

Comment 1.5: The experiments don't include direct comparison with other methods to quantitatively show why the proposed method is better.

Response 1.5: We have performed a comparison on the accuracy of our model with that in the reported literature and included it as a table.  A direct quantitative comparison in the method of using certainty measure to assess the heatmap and information content in the signal could not be performed as we could not identify any similar studies performed in the past. We have incorporated a comparison with similar literatures, which used different approaches in the assessment of confidence in the models.

Comment 1.6: The manuscript is in general well written, but the introduction and the related work need more work.

Response 1.6: We have modified both introduction and related work section to improve readability and clarify the results.

Reviewer 2 Report

Review for manuscript Entropy-2246293-v1 “Informational entropy measures for evaluation of reliability of deep neural network results” by E. Gireesh and V. Gurupur.

This manuscript reports a study of the decision-making certainty and confidence of Deep Neural Networks (DNN) and how this relates to the information content of classified data. The authors achieve this be creating a novel certainty index which is applied to the outputs of the most penultimate layer of the DNN. The authors test their certainty index with DNNs trained and tested on iEEG data and relate the index to the data’s sample entropy. The authors also systematically explore the effects of various parameters (e.g. sample length, sample entropy embedding dimension and scaling parameter) on this relationship. The authors found a stronger relationship between the certainty index and the sample entropy than between the certainty index and the actual signal used to train the DNN. The authors conclude that decision certainty of a DNN is related to the network’s ability to capture the information in original signal with the DNN focusing on information rich portions of as signal to arrive at decisions.

            I found this study to propose an interesting index of DNN decision-making confidence that may have many useful research applications. Moreover, the finding that the DNN decision certainty is more greatly related to the information in a signal that the raw signal values may be helpful for our understanding of DNN behavior and the nature of information more generally. The experimental and analytical methods used in this study appear to be well-implemented and well-reported for the most part (but see comments below). However, the English writing and grammar needs to be substantially improved, along with a few additional issues (listed below) that, if addressed, will improve the strength of the authors’ conclusions, as well as the readability of the manuscript.

p.1, lines 19 – 31: The authors introduce a couple of acronyms (EEG, iEEG) here without their definitions. All acronyms should be defined in the manuscript.

p.2, lines 40 – 42: The authors state “Subsequently Sample entropy was introduced which excludes the self-matches. This measure does not depend on the signal length and has higher relative consistency”. If the sample entropy does not depend on signal length, why was signal length one of the parameters systematically explored in this study?

p.3 lines 51 – 53: The authors state “Once a model is trained the reliability of information capturing capacity of DNN can be assessed by evaluating heatmaps, which help in explaining what the DNN has identified as significant in arriving at a decision”. Can the authors provide a more direct and explicit definition of a heatmap in this context?

p.3, Section 2.1, lines 106 - 113: can the authors provide a motivation for the definition of certainty given in Equation 1? It seems this equation is essentially expressing an unnormalized deviation of one output unit relative to the mean of the other units. How does this express certainty in the present context? Also, the authors should explicitly define the variable y, i.e. that it reflects the activity of a given unit, to make things clear to the reader.

p.3, Section 2.2, lines 114 - 126: Just to be clear, the variable k here indexes the different features used by the DNN (defined by a 2D map indexed by i and j) to classify categories c, correct? Again, it is helpful to the reader to have all variable explicitly defined.

p.4, lines 127 - 129: It is unclear which variables x and y refer to the heatmap and signal in Equation 4. Also, why doesn’t Equation 3 incorporate the relevant heatmap variable used in Equation 4 in its definition? For ease of reader understanding, variables should be consistent across all mathematical formula.

p.4, lines 129 – 132: Here the variable x is used to represent the data in Equation 5. Does this mean that variable y represents the heatmap in Equation 4? If so, then what does y represent in Equation 1? Again, variables should be consistent across all mathematical formula.

p.5, lines 158 – 160: The authors state “The model was trained with 9000 samples and tested with 1000 samples. The accuracy in the model is depicted in the confusion matrix as noted in the figure”. Was any kind of cross-validation performed on the model across different divisions of the data set?

Author Response

Comments and Suggestions for Authors

Response to Reviewer 2

Review for manuscript Entropy-2246293-v1 “Informational entropy measures for evaluation of reliability of deep neural network results” by E. Gireesh and V. Gurupur.

Comment 2.1: This manuscript reports a study of the decision-making certainty and confidence of Deep Neural Networks (DNN) and how this relates to the information content of classified data. The authors achieve this be creating a novel certainty index which is applied to the outputs of the most penultimate layer of the DNN. The authors test their certainty index with DNNs trained and tested on iEEG data and relate the index to the data’s sample entropy. The authors also systematically explore the effects of various parameters (e.g. sample length, sample entropy embedding dimension and scaling parameter) on this relationship. The authors found a stronger relationship between the certainty index and the sample entropy than between the certainty index and the actual signal used to train the DNN. The authors conclude that decision certainty of a DNN is related to the network’s ability to capture the information in original signal with the DNN focusing on information rich portions of as signal to arrive at decisions. I found this study to propose an interesting index of DNN decision-making confidence that may have many useful research applications. Moreover, the finding that the DNN decision certainty is more greatly related to the information in a signal that the raw signal values may be helpful for our understanding of DNN behavior and the nature of information more generally. The experimental and analytical methods used in this study appear to be well-implemented and well-reported for the most part (but see comments below). However, the English writing and grammar needs to be substantially improved, along with a few additional issues (listed below) that, if addressed, will improve the strength of the authors’ conclusions, as well as the readability of the manuscript.

Response 2.1: Thank you very much for the detailed and valuable comments, which have significantly helped in improving the manuscript. We have made the appropriate changes to the main text.

Comment 2.1: p.1, lines 19 – 31: The authors introduce a couple of acronyms (EEG, iEEG) here without their definitions. All acronyms should be defined in the manuscript.

Response 2.1: We have expanded these acronyms in their first appearance in the text

Comment 2.3: p.2, lines 40 – 42: The authors state “Subsequently Sample entropy was introduced which excludes the self-matches. This measure does not depend on the signal length and has higher relative consistency”. If the sample entropy does not depend on signal length, why was signal length one of the parameters systematically explored in this study?

Response 2.3: We were trying to convey that the sample entropy is largely independent of the length of the time series as reported in the previous literature (Physiological time-series analysis using approximate entropy and sample entropy (physiology.org), fig 2).  They do report that there is some level of variation with the sample length, which is analyzed in other studies also.  That was the reason for us exploring this variability in this study. We have modified the text in this section to indicate that the role of exploring the parameter space.    

Comment 2.4: p.3 lines 51 – 53: The authors state “Once a model is trained the reliability of information capturing capacity of DNN can be assessed by evaluating heatmaps, which help in explaining what the DNN has identified as significant in arriving at a decision”. Can the authors provide a more direct and explicit definition of a heatmap in this context?

Response 2.4: We have elaborated on the direct definition of heatmap and explained the specific application pertaining to the iEEG signal (lines 50-58).

Comment 2.5: p.3, Section 2.1, lines 106 - 113: can the authors provide a motivation for the definition of certainty given in Equation 1? It seems this equation is essentially expressing an unnormalized deviation of one output unit relative to the mean of the other units. How does this express certainty in the present context? Also, the authors should explicitly define the variable y, i.e. that it reflects the activity of a given unit, to make things clear to the reader.

Response 2.5: We have explained the motivation for using this measure from the standpoint of biological neural network and from the standpoint of application along with grad-CAM algorithm (This is included in the introduction section: Lines after 70). We have indicated that for comparison across layers the certainty value can be normalized with standard deviation. The variable y, certainty index c and number of nodes N is explained in detail after the equation for certainty. Given the fact that the final layer has only two elements, based on this definition the certainty index will be  yi- yj  which we have explained the last part of the section on certainty index.

Comment 2.6: p.3, Section 2.2, lines 114 - 126: Just to be clear, the variable k here indexes the different features used by the DNN (defined by a 2D map indexed by i and j) to classify categories c, correct? Again, it is helpful to the reader to have all variable explicitly defined.

Response 2.6: Yes, the “k” here indexes the different activation maps, which essentially means the different features used by DNN. Please note that the feature maps are one dimensional in this case and the individual elements within the feature map are indexed by “i”.  That is why index “j ” is not included, which is correction from the original submission. This is explicitly included in the text.

Comment 2.7: p.4, lines 127 - 129: It is unclear which variables x and y refer to the heatmap and signal in Equation 4. Also, why doesn’t Equation 3 incorporate the relevant heatmap variable used in Equation 4 in its definition? For ease of reader understanding, variables should be consistent across all mathematical formula.

Response 2.7: We have modified the equation 4 to give better clarity among other equations.  This directly incorporated the symbol for the heatmap (gc) as obtained from equation 3 and the symbol for data (x) as used in equation 5 and in the explanation prior to equation 6.

Comment 2.8: p.4, lines 129 – 132: Here the variable x is used to represent the data in Equation 5. Does this mean that variable y represents the heatmap in Equation 4? If so, then what does y represent in Equation 1? Again, variables should be consistent across all mathematical formula.

Response 2.8: As mentioned in the previous responses we have clarified the symbols with uniformity across different equations. The variable x in equation 5 represent data (similar to that in equation 4 and in the explanation prior to equation 6).

Comment 2.9: p.5, lines 158 – 160: The authors state “The model was trained with 9000 samples and tested with 1000 samples. The accuracy in the model is depicted in the confusion matrix as noted in the figure”. Was any kind of cross-validation performed on the model across different divisions of the data set?

Response 2.9: A 10 fold cross validation was performed as added in this section. Also, as suggested by the other reviewer we have included additional data from 7 patients, along with cross validation of that data.  

Round 2

Reviewer 1 Report

The authors revised the manuscript according to my previous comments.

Reviewer 2 Report

The authors have satisfactorily addressed all of my concerns.